

# STB: synthetic minority oversampling technique for tree-boosting models for imbalanced datasets of intrusion detection systems

Li-Hua Li, Ramli Ahmad, Radius Tanone and Alok Kumar Sharma

Information Management, Chaoyang University of Technology, Taichung, Wufeng District, Taiwan

## ABSTRACT

Attacks on the Intrusion Detection System (IDS) can result in an imbalanced dataset, making it difficult to predict what types of attacks will occur. A novel method called SMOTE Tree Boosting (STB) is proposed to generate synthetic tabular data from imbalanced datasets using the Synthetic Minority Oversampling Technique (SMOTE) method. In this experiment, multiple datasets were used along with three boosting-based machine learning algorithms (LightGBM, XGBoost, and CatBoost). Our results show that using SMOTE improves the content accuracy of the LightGBM and XGBoost algorithms. Using SMOTE also helps to better predict computational processes. proven by its accuracy and F1 score, which average 99%, which is higher than several previous studies attempting to solve the same problem known as imbalanced IDS datasets. Based on an analysis of the three IDS datasets, the average computation time required for the LightGBM model is 2.29 seconds, 11.58 seconds for the XGBoost model, and 52.9 seconds for the CatBoost model. This shows that our proposed model is able to process data quickly.

## INTRODUCTION

Internet-connected devices (*Xu et al., 2023*) such as smartphones and laptops are increasingly being used to exchange information, making network security crucial. Network administrators (*Jemmali et al., 2022*) need to prepare for offline and online data exchange on a secure computer network. Although there are several security methods (*Shah et al., 2023*), attacks can still disrupt network functions. To protect information passing through the network, it is important to keep it secure and confidential.

Intrusion Detection System (IDS) (*Ghanem et al., 2022*) is an essential part of a network security system that monitors network activity to detect and prevent attacks. IDS can identify attack patterns in network packets (*Raharjo et al., 2022*), monitor user behavior and detect abnormal traffic. However, no network is completely secure (*El Houda, Brik & Khoukhi, 2022*), and optimal IDS performance is required to immediately discover and

Corresponding author
Ramli Ahmad,
s10814904@gm.cyut.edu.tw

secure the network. Due to the limited human ability to detect various attacks, information technologies such as Artificial Intelligence and machine learning should be used.

Machine learning research is constantly evolving, bringing with it new methods and models. A good dataset is critical for accurate machine learning predictions, but in fact, IDS surveillance records produce imbalanced data. Imbalanced datasets (*Wang et al., 2022*) and missing datasets can affect the accuracy of machine learning in classifying attacks. To solve this problem, the Synthetic Minority Oversampling Technique (SMOTE) (*Das et al., 2020*) method is used to generate the data (*Liu et al., 2021*).

The SMOTE method is one of the oversampling techniques used in unbalanced data processing, especially in classification. The advantage of SMOTE compared to other oversampling methods is that it generates synthetic data for minority classes by randomly selecting two or more similar samples and combining them. In this way, SMOTE creates new samples that still reflect the features and patterns of minority classes, thus maintaining the integrity of the information on the synthetic data. Simple oversampling methods, such as data replication or simple duplication, can lead to overfitting, where models tend to remember training data very well but cannot generalize well to new data. SMOTE helps address this problem by creating variations within minority classes, which prevents models from overfitting on the same data.

The novelty of this research is to develop a new method called STB (SMOTE Tree Boosting) to solve the problem of imbalanced IDS datasets using the SMOTE method. The research focuses on the proposed SMOTE technique to improve machine learning performance of Tree Boosting (LightGBM, XGBoost and CatBoost) called STB to solve imbalanced IDS datasets namely: KDDCup99, CICIDS 2017 and UNSW_NB15. Then we compare the outcomes to previous research (*Lin et al., 2022*) with the same problem.

In addition, the Related Work section will address literature reviews. IDS methods for imbalanced data are discussed in the Proposed Method section. Experimental results are discussed in the Results section and the rest is the conclusion of the study.

## RELATED WORK

Recently, researchers still use IDS imbalanced datasets in their research including KDDCup99 (*Madhavi & Nethravathi, 2022*), CICIDS 2017 (*Leon, Markovic & Punnekkat, 2022*), dan UNSW NB 15 (*Shukla et al., 2023*). *Madhavi & Nethravathi (2022)* proposed the development of an intrusion detection model based on a combination of Gradient Boosted Decision Tree (GBDT) and Gray Wolf Optimization (GWO). GBDT is used as a powerful learning algorithm while GWO is used to optimize parameters in the model. With this approach, the study aims to improve the performance and accuracy of intrusion detection in the face of unbalanced data. It is expected that the resulting model can be effective in detecting suspicious network activity or threats related to network security. However, only one dataset, KDDCup99, was used in their research, so their proposed model had to be retested with a different IDS dataset to test the reliability of their proposed model. From the experimental result, their accuracy value is 96.20% and their F1 score value is 95.81%. These results still need to be improved.

Another study (*Leon, Markovic & Punnekkat, 2022*) conducted comparative evaluations of different machine learning algorithms for network intrusion detection and attack classification. They used four unbalanced IDS data sets, namely KDDCup99, NSL-KDD, UNSW NB15 and CIC-IDS-2017. In their research study, the researchers used various machine learning algorithms that detect network intrusions. They analyzed and compared the performance of several popular algorithms such as decision tree, naive Bayes, support vector machines, neural networks or other algorithms commonly used in intruder detection. However, their experimental results show only accuracy comparisons and no F1 score results for each of their machine learning algorithms, and the accuracy values for the UNSW NB15 and CICIDS 2017 datasets are still below 90%. These results still need to be improved.

In addition, *Shukla et al. (2023)* uses UNSW-NB15 to predict attacks on IDS imbalanced dataset. Researchers developed UInDeSI4.0, an efficient unattended intrusion detection system for network traffic flow in the industry 4.0 ecosystem. This system can detect suspicious or unauthorized activity in computer networks without requiring prior training data. This research focuses on intrusion detection in network traffic flows in the context of Industry 4.0 ecosystems, and includes the use of digital technologies, advanced automation, and machine-to-machine communication to improve efficiency and productivity in the manufacturing industry. However, the accuracy of the model used is below 70% and the f1 score is not displayed. These results still need to be improved.

Our research certainly focuses on three types of datasets, but we discovered that some datasets are not balanced. It will be interesting to see how this imbalanced dataset affects us in the future. We're attempting to balance a dataset using SMOTE. However, we are aware that other researchers have used SMOTE in their work. *Alshamy et al. (2021)* using the SMOTE Technique to deal with a class imbalanced problem and RF classifier that has improved performance to detect types of attack. Furthermore, they use several combinations of algorithms to classify and compare the performance of each algorithm used. It's the same as *Rani & Gagandeep (2022)* who used SMOTE in their research where SMOTE was used as a data oversampling technique to balance the dataset.

In contrast to previous studies that used many open datasets, our study processes SMOTE to deal with imbalanced datasets and employs the LightGBM algorithm. The goal is to improve the performance of the boosting algorithm we use. Of course, other researchers have independently used SMOTE and algorithms such as LightGBM. However, in this study, we wanted to use it and found better results. The split data will then be used for training using machine learning algorithms using the SMOTE approach. Additionally, the results before and after SMOTE will be compared to the use of other boosting algorithms such as XGBoost and CatBoost.

Given this situation, a machine learning intervention on IDS as part of the AI is required to help identify the existing attack types. In addition, advances in IDS relate to the types of attacks that are common today. We developed this study to address the problem of identifying the types of network attacks. We identify several types of attacks: analysis, backdoor, shellcode, worms and infiltration. Furthermore, this study aims to identify the types of attacks that exist on IDS using a machine learning algorithm approach. The

following are the specific objectives of the study: A novel method called STB is proposed to generate synthetic tabular data from imbalanced datasets using the SMOTE method. Moreover, we compared three well-known boosting algorithms, LightGBM, XGBoost, and CatBoost, to see how SMOTE affected the dataset we used both in terms of F1 scores as well as accuracy and processing speed.

## PROPOSED METHOD

This chapter discusses the materials and methods used to improve the accuracy prediction performance of IDS based on imbalanced datasets using SMOTE. As for more than one dataset that is the same that we use to see the existing improvisations. In this chapter, we have prepared a flowchart of the proposed method, which consists of dataset preparation to model performance evaluation and comparisons for each dataset used.

In conducting this research, we designed the preparation of the dataset to produce the predicted output in the flowchart of the proposed method as shown in Fig. 1.

Figure 1 illustrates the flow of our proposed method, starting with the preparation of a dataset consisting of 3 types of datasets named KDDCUP99 with 4 attack categories, CICIDS 2017 with 6 attack categories and UNSW NB15 with 9 attack categories attacks. Next stage is pre-processing which consists of three stages, namely data cleaning, normalization, and label encoding. Because the amount of data is imbalanced, SMOTE is needed to balance the dataset. From here, then the dataset will be split into two 80% for training and 20% for testing. Next step is to build LightGBM, XGBoost, CatBoost models to make predictions. After the prediction process is complete, the next step is to create an evaluation matrix to determine the accuracy of each model for the three datasets. The final step is to collect the accuracy, precision, recall and F1 score results from each model for the three datasets and compare one model to another model to find the best model and the impact of SMOTE on the ability of the three machine learning models to figure out accurate prediction.

### Experiment

In this experiment, the dataset that we use is an open dataset, which in detail describes the real condition of a network that has an IDS. The field in the dataset with the name KDDCup99 is available at the link: (https://www.kaggle.com/datasets/galaxyh/kdd-cup-1999-data), CICIDS 2017 is available at the link: (https://www.unb.ca/cic/datasets/ids-2017.html), and UNSW NB15 is available at the link: (https://research.unsw.edu.au/projects/unsw-nb15-dataset), describes the conditions of the attack which we mapped into parts of the attack on IDS. As for this dataset, we make dataset labels as in Tables 1, 2 and 3. We provide labels for each attack name at the label encoding stage in the pre-processing process. Data pre-processing is the initial technique of machine learning and data mining to convert raw data or commonly referred to as raw data collected from various sources into cleaner information and use it for further processing. This process can also be described as the first step in gathering all available information by cleaning, filtering, and combining

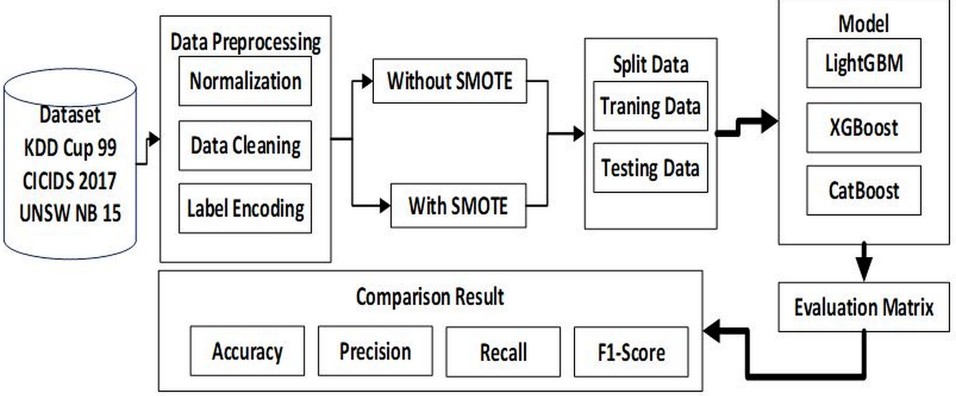

**Figure 1** Flowchart proposed method.

**Table 1 KDDCup99 dataset.**

| Attack Name | Label | Data | Without SMOTE | | With SMOTE | |
|---|---|---|---|---|---|---|
| | | | Training | Testing | Training | Testing |
| Normal | 0 | 97,277 | 77,852 | 19,425 | 77,702 | 19,575 |
| Probe | 1 | 4,107 | 3,310 | 797 | 3,283 | 824 |
| DoS | 2 | 391,458 | 313,109 | 78,349 | 313,273 | 78,185 |
| U2L | 3 | 52 | 37 | 15 | 800 | 200 |
| R2L | 4 | 1,126 | 908 | 218 | 918 | 208 |

**Table 2 CICIDS 2017 dataset.**

| Attack Name | Label | Data | Without SMOTE | | With SMOTE | |
|---|---|---|---|---|---|---|
| | | | Training | Testing | Training | Testing |
| BENIGN | 0 | 22,731 | 18,257 | 4,474 | 18,129 | 4,602 |
| Bot | 1 | 1,966 | 1,568 | 398 | 1,584 | 382 |
| BruteForce | 2 | 2,767 | 2,226 | 541 | 2,243 | 524 |
| DoS | 3 | 19,035 | 15,202 | 3,833 | 15,225 | 3,810 |
| Infiltration | 4 | 36 | 28 | 8 | 1,000 | 240 |
| PortScan | 5 | 7,946 | 6,328 | 1,618 | 6,382 | 1,564 |
| WebAttack | 6 | 2,180 | 1,719 | 461 | 1,736 | 444 |

this data. Data pre-processing is very important as errors, redundancies, missing values, and inconsistent data lead to reduced accuracy of analysis results.

Tables 1, 2 and 3 demonstrate the description of the dataset used in this experiment. In the original dataset, it appears that there is an imbalanced dataset that occurs. To deal with this problem, SMOTE is used and the number of datasets becomes balanced. Furthermore, each dataset is split into several training and testing datasets with a proportion of 80:20.

**Table 3  UNSW NB15 dataset.**

| Attack Name | Label | Data | Without SMOTE | | With SMOTE | |
|---|---|---|---|---|---|---|
| | | | Training | Testing | Training | Testing |
| Normal | 0 | 37,000 | 29,616 | 7,384 | 29,616 | 7,384 |
| Analysis | 1 | 677 | 538 | 139 | 1,000 | 139 |
| Backdoor | 2 | 583 | 466 | 117 | 1,000 | 117 |
| DoS | 3 | 4,089 | 3,300 | 789 | 3,300 | 789 |
| Exploits | 4 | 11,132 | 8,865 | 2,267 | 8,865 | 2,267 |
| Fuzzers | 5 | 6,062 | 4,824 | 1,238 | 4,824 | 1,238 |
| Generic | 6 | 18,871 | 5,122 | 3,749 | 15,122 | 3,749 |
| Reconnaissance | 7 | 3,496 | 2,803 | 693 | 2,803 | 693 |
| Shellcode | 8 | 378 | 295 | 83 | 1,000 | 100 |
| Worms | 9 | 44 | 36 | 8 | 1,000 | 100 |

## Normalization

Data normalization is the basic element of machine learning and data mining to ensure the record stays consistent in the data set. In this study, we changed the name of the column to a uniform form of initialization, such as the columns \"Flow Duration\", \"Total Fwd Packets\", \"Total Backward Packets\", etc., we changed it to F1, F2, F3, etc. In the normalization process, data transformation is required or converts original data into a format that allows for efficient data processing. The main purpose of data normalization is to eliminate data redundancy (repetition) and standardize information for better data workflows. Data normalization is used to spread an attribute's data to fall within a smaller range, *e.g.*, −1 to 1 or 0 to 1. This is generally useful for classification algorithms. Data normalization techniques are very helpful as they offer many advantages as follows:

- The application of machine learning and data mining algorithms is easier
- Machine learning and data mining algorithms become more effective and efficient
- Data can be extracted from the database faster
- Normalized data can be analyzed by specific methods

## Data cleaning

Data cleansing or data cleansing is one of the steps in data pre-processing. The purpose of this cleansing data is to select and eliminate data that has the potential to reduce the accuracy of machine learning and artificial intelligence. At this stage, we need to overcome the problematic data on our IDS dataset. In this study, some values are missing from the KDD Cup 99 dataset, so we manually filled in the missing data by averaging between above and below the missing value data. With the exception of the CICIDS 2017 and UNSW NB15 datasets, these datasets do not have a missing value. Some common problems encountered in datasets are as follows:

- Missing value when a value is missing from the record. For example, in a row table's data, there is a cell with no value. To get around this, we replace the missing value with

the median method. For this purpose, the mean value is determined and missing values are replaced by the mean value.

- Noisy data when the data contains incorrect or anomalous values. The condition is also known as an outlier. In this study the KDD Cup 99, CICIDS 2017, and UNSW NB15 datasets had noisy data, these datasets had problems with inaccurate labeling and a large number of duplicates. To overcome noisy data, there are several techniques that we performed, including:

  - Binning, a method we use that divides data into multiple partitions and then treats the partitions separately. Then the mean, median or specified limit value is searched from all data partitions.
  - Regression, a method we use that uses the linear regression equation to predict the value of data. This method can be used when there is only one independent attribute.
  - Clustering, a method we use to create a group or cluster of data of similar value. Values that don't get into the cluster can be considered noisy data.

- Inconsistent data, which is the condition when the values in the data are inconsistent. In this study, the data from KDD Cup 99, CICIDS 2017 and UNSW NB 15 showed inconsistent data. To overcome this inconsistent data, we use noisy data, binning, regression and clustering methods.

## Label encoding

Label encoding is a pre-processing of IDS dataset where we try to change the data type of the categorical column to numeric (from string to numeric). The columns we changed include: For the KDD Cup 99 record, the records in the Protocol_type, service, and Flag columns have string values like tcp, http, sf, etc. We change them to the numbers 1, 2, 3, etc. For UNSW NB 15 records, in the Protocol, Service, and Status columns, the records have string values like tcp, ftp, and FIN, etc., we change them to the numbers 1, 2, 3, etc. Meanwhile contains the CICIDS 2017 dataset doesn't have a string value in each dataset, so we won't change anything. This happens because the machine learning model does not understand the character of the string, and therefore a provision must be made to convey it in a format that the machine learning can understand. This is achieved through the coding label method. In the tagging coding method, the category under the categorical feature is changed in a manner involving hierarchical separation. That is, if we have categorical features in which the categorical variables are hierarchically related to each other, then we need to label those features. When label-encoding is performed on non-hierarchical features, the accuracy of the model is severely compromised and is therefore not a good choice for non-hierarchical features. To see more detailed dataset labels, please refer to Tables 1, 2 and 3.

## SMOTE for imbalanced dataset

Imbalanced data is a situation where the target class or category has a significantly different frequency in the training data. This means that one or more classes are under-represented in the data set, while one or more classes are over-represented. In this study, the datasets we used had imbalanced classes, including: In the KDD Cup 99 dataset, some classes in

the dataset had a much larger number of samples than others, with the class representing the attack having a much smaller number of samples compared to normal traffic classes. In the CICIDS 2017 and UNSW NB 15 datasets, some classes have attack frequencies, anomalous representation, and consistency that differ from normal classes. This data set also contains duplicate and irrelevant data. Imbalanced data can cause problems with machine learning models as they can be biased toward the majority class and have trouble spotting the minority class. This can lead to poor performance metrics for minority classes, such as accuracy, precision recall and F1 Score. To overcome this problem, the researcher proposes first determining the nearest majority data point from the sample selection and then replacing the nearest neighbour parameter with a safe radius distance. Then the safety radius becomes a reference in creating synthetic stone data points. The circle formula with a two-dimensional vector is used to generate data as shown in formula (1).

$$\| \underset{\sim}{b} - p \| \le r^2 \tag{1}$$

$$\sum_{i=1}^{n} \left( b_{ij} - P_{ij} \right)^2 \le r^2 \tag{2}$$

$$r^2 = \sum_{j=1}^{n} \left( p_j - t_j \right)^2 \tag{3}$$

Where the point "$p$" is the minority sample point which is the center of the circle with $(p_1, p_2, p_3, \ldots, p_n)$ as in formula (2). Point "$b$" is a new generation synthetic point of interpolation between the two directions with $(b_1, b_2, b_3, \ldots, b_n)$ as in formula (2). Point "$t$" is the majority point closest to the center of the circle with $(t_1, t_2, t_3, \ldots, t_n)$ as in formula (3). On the other hand, $r^2$ is the distance between $p$ and $t$ can in formula (3).

The next step is to perform calculations to generate new synthetic data. Using Euclidean formula, the distance between each majority data and each minority sample is calculated. The smallest distance from the entire selected minority sample is the majority data point. As shown in formula (4).

$$r_{ij} = min \sum_{i=1}^{n} \sum_{j=1}^{n} \sqrt{\left( p_j - t_i \right)^2} \tag{4}$$

where $r_{ij}$ is the smallest distance between majority sample $i$ and minority sample $j$. The generation of new synthetic data is carried out on the interpolation line between the next predetermined majority data points and the minority data sample points. The new generation of synthetic data based on the safe radius distance from the first direction interpolation scheme is categorized as $a_i$ with $(a_1, a_2, a_3, \ldots, b_n)$. In addition, both line directions are used to generate synthetic $r_{ij}$ and $-r_{ij}$, which can be seen in formula (5) and (6).

$$a_{ij} = P_j + \left( rand0, 1 \times \left( r_{ij} - p_j \right) \right) \tag{5}$$
$$b_{ij} = P_j + \left( rand0, 1 \times \left( p_j - r_{ij} \right) \right). \tag{6}$$

The restricted area for creating new synthetic data is then used to reduce the occurrence of overlapping data in the SMOTE process.

The Synthetic Minority Oversampling Technique (SMOTE),

Algorithm SMOTE (S, M, A)

Input:

$A$ = Number of nearest neighbors

$M$ = Number of samples from minority classes

$S$ = Amount of SMOTE $S$ %

**Output:** $(\frac{S}{100})$ * $M$ Samples of synthetic minority classes

("Identify minority class samples in the data set. If S is less than 100%, randomize the

minority class samples since only a random percentage of them will be SMOTE")

**if** $S < 100$

 **Then** determine the minority class sample M randomly

M = $(\frac{S}{100})$ *$M$

$S = 100$

endif

$S = (\text{int})(\frac{S}{100})$ ("determine the number of SMOTE samples assuming that the number

is an integer with a multiple of 100")

$A$ = Number of nearest neighbors

$numattrs$ = Number of attributes

*Sample* [ ] [ ]: is a number of original samples in the form of Arrays from the minority

class

*newindex*: the resulting number of samples is calculated by initializing to 0

*Synthetic* [ ] [ ]: Array for synthetic samples ("only calculates the number of nearest

neighbors of A for each sample")

for $i \leftarrow 1$ **to** $M$

nearest neighbor i is calculated using A then store index into array n

Populate ($S$, $i$, $n$ array)

endfor

*Populate* ($S$, $i$, $nnarray$) ("this is a manufacturing function with synthetic samples")

**while** $S \neq 0$

select data one from "A" for nearest neighbor "i" by randomly selecting 1 and A

for $attr \leftarrow 1$ **to** $numattrs$

Compute: $dif = Sample\,[nnarray\,[nn\,]]\,[attr\,] - Sample\,[i\,][attr\,]$

Compute: *gap* = random number between 0 and 1
*Synthetic* [*newindex* ][*attr* ] = *Sample* [*i* ][*attr* ] + *gap* *
*dif*
**endfor**
*newindex* ++
*S* = *S* − 1

endwhile
**return** (∗ End of Populate. ∗)

## Splitting data

In this study, the first dataset used was KDDCup99 with a total number of records are 494,020, then the CICIDS 2017 dataset with a total number of records are 56,661 and finally UNSW NB15 with a total number of records are 540,044. With this split data, the data from each dataset is divided into two, namely training data and test data, with a ratio of 80% for training data and 20% for test data. Split data is a more efficient option because it can split Intrusion Detection System (IDS) datasets into training subsets and test subsets quickly compared to cross validation because it will take a long time. The use of cross validation will cause the results of model evaluation to vary depending on how the data is divided into different subsets. This will lead to a degree of dependence on the data sharing performed. We want to avoid this dependency and use the data split method to get more consistent results. After dividing the two data into training data and test data, the amount of data is: For the KDD Cup 99 dataset, the number of training data is 395,216 and the test data is 98,804. For CICIDS 2017, the number of training data is 45,328 and the test data is 11,333. For the UNSW NB 15 dataset, the training dataset is 108,009 and the test dataset is 432,035.

## Apply the machine learning models

In this research, we use three-boosting machine learning models including: LightGBM, XGBoost and CatBoost as explained in the previous chapter. These three-boosting machine learning has its own unique characteristics and learning approach. By using three different algorithms, we introduce diversity in the models being trained. This helps to capture different aspects of the imbalanced dataset and potential patterns associated with intrusive instances. The combination of multiple models can lead to more robust and accurate predictions. Parameters we use in each model which can be seen below.

## LightGBM

LightGBM is a gradient boosting framework that is efficient and distributed, providing faster training speed, greater efficiency, less memory usage, and better accuracy. It is a decision tree-based machine learning algorithm that competes with XGBoost, which used to dominate Kaggle competitions. LightGBM is now more popular among Kagglers because of its high speed, accuracy, and ability to handle large amounts of data. It is also known for its GPU learning support and its focus on accuracy. However, LightGBM is not advisable for small datasets and is sensitive to overfitting. In this research of the LightGBM model,

the number of leaves per tree we use is 63, the speed iteration control is 0.01, specifies the fraction of data to use for each iteration, and is generally used to guide training speed up and avoid overfitting with a value of 0.5. The proportion of features is randomly chosen in each iteration to create trees with a value of 0.5.

## XGBoost

XGBoost is a machine learning algorithm used for regression, classification and ranking problems. It's open source and efficient for large and complex datasets. XGBoost can handle missing values and function interactions, has built-in regularization techniques, and works by building a series of decision trees that correct the errors of previous trees. A popular choice among data scientists, XGBoost is used in various applications such as financial modelling, healthcare, and natural language processing. In the XGBoost model, n_estimators, which determine the epoch of the model, *is* set to 100 and early_stopping_rounds to 10 to check for overfitting. The grid values sought for the learning rate *is* 0.01. The max_depth is 4 and the min_child_weight in selected ranges from 1 to 10.

## CatBoost

CatBoost is an open-source gradient boosting library used for classification, regression, and ranking tasks. It handles categorical variables better than other algorithms, making it useful for datasets with a mix of numeric and categorical data. It uses ordered boosting to process categorical variables and handles categorical variables with high cardinality more effectively. CatBoost can handle missing values without any special pre-processing, saving time and effort for data scientists. It is powerful, efficient and used in various applications like web search ranking, recommender systems and computer vision. In the CatBoost, the grid values sought for the learning rate is 0.01. The iteration is 100 and the maximum depth of the tree is 10. It is capable of handling model overfitting and the number of l2_leaf_reg per tree we use is 1.

## Evaluation metrics

To evaluate the performance of LightGBM, XGBoost, and CatBoost on the three datasets we use, the metrics we use are as follows:

$$\text{Precision} = TP/(TP + FP) \tag{7}$$

$$\text{Recall} = TP/(TP + FN) \tag{8}$$

$$\text{Accuracy} = (TP + TN)/(TP + TN + FP + FN) \tag{9}$$

$$\text{F1-Score} = (2 \times \text{Precision} \times \text{Recall})/(\text{Precision} + \text{Recall}). \tag{10}$$

Precision is the ratio of positive correct predictions compared to the overall positive predicted results in the IDS dataset used. Recall is the ratio of correctly positive predictions compared to all the correctly positive data. Is the ratio of correct predictions (positive and negative) with all the IDS data that we use. The F1-score is a comparison of the average precision and recall which is weighted. Finally, accuracy answers the question "how

**Table 4  Performance of 3 models with and without SMOTE.**

| Model | Evaluation | KDDCup99 | | CICIDS 2017 | | UNSW_NB15 | |
|-------|-----------|----------|---|-------------|---|-----------|---|
| | | Without SMOTE | With SMOTE | Without SMOTE | With SMOTE | Without SMOTE | With SMOTE |
| LightGBM | accuracy | 0.878 | 0.977 | 0.653 | 0.996 | 0.845 | 0.893 |
| | Precision | 0.920 | 0.975 | 0.721 | 0.996 | 0.857 | 0.899 |
| | Recall | 0.878 | 0.977 | 0.653 | 0.996 | 0.845 | 0.893 |
| | F1-Score | 0.887 | 0.976 | 0.657 | 0.996 | 0.850 | 0.895 |
| XGBoost | accuracy | 0.878 | 0.999 | 0.998 | 0.995 | 0.899 | 0.893 |
| | Precision | 0.920 | 0.999 | 0.998 | 0.995 | 0.901 | 0.901 |
| | Recall | 0.878 | 0.999 | 0.998 | 0.995 | 0.899 | 0.893 |
| | F1-Score | 0.887 | 0.999 | 0.998 | 0.994 | 0.897 | 0.896 |
| CatBoost | accuracy | 1.000 | 0.999 | 0.997 | 0.996 | 0.892 | 0.887 |
| | Precision | 1.000 | 0.999 | 0.997 | 0.996 | 0.893 | 0.889 |
| | Recall | 1.000 | 0.999 | 0.997 | 0.996 | 0.892 | 0.887 |
| | F1-Score | 1.000 | 0.999 | 0.997 | 0.996 | 0.888 | 0.887 |

accurately are the predictions of the types of attacks processed by LightGBM, XGBoost, and CatBoost".

# RESULTS

In Table 4, we present the experimental results obtained with SMOTE on LightGBM. We also compare these results to those of other boosting algorithms, such as XGBoost and CatBoost, to see how the algorithm performs when SMOTE is used.

Next, to see the performance comparison of the three types of boost algorithms, we present them in graphical form in Figs. 2–4.

Figure 2 shows that when the KDDCup99 dataset was oversampled using SMOTE, the results improved. This is evident from the tables and graphs. The accuracy of the LightGBM model without SMOTE is 0.878, and the accuracy after SMOTE is 0.977. This demonstrates that SMOTE can improve the LightGBM model's capability by 0.098, or 9.8%. The accuracy obtained before using SMOTE in the 2017 CICIDS dataset was 0.653, and the accuracy obtained after using SMOTE was 0.996. This demonstrates that the SMOTE oversampling method can improve the performance of the LightGBM model by 0.343, or 34.3%, in the 2017 CICIDS data. The accuracy obtained without using SMOTE in the UNSW NB15 dataset is 0.845, and the accuracy obtained after using SMOTE is 0.893, demonstrating that there is a 0.048 or 4.8% improvement in performance in the LightGBM model.

Furthermore, based on the accuracy of the LightGBM model on the data from the three IDS datasets mentioned above, it appears that the oversampling method using SMOTE can improve the performance of the LightGBM model, which can also be attributed to increasing precision, recall, and F1-score on the LightGBM model when processing the three IDS datasets.

Figure 3 demonstrated that in the KDDCup99 dataset, the accuracy obtained on the XGBoost model before SMOTE was 0.878, and the accuracy obtained after SMOTE was 0.999, demonstrating that the performance of the XGBoost model improved by 0.121, or

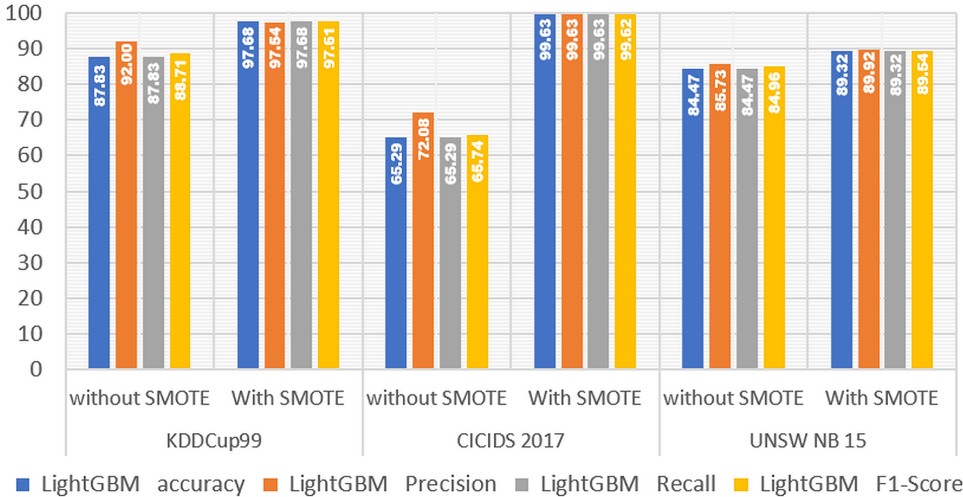

**Figure 2** LightGBM performance without and with SMOTE.

12.1%. In the 2017 CICIDS dataset, using SMOTE reduced XGBoost performance slightly but not significantly. The accuracy obtained by the XGBoost model without SMOTE was 0.998, while the accuracy obtained after using SMOTE was 0.995. This demonstrates that the performance of the XGBoost model decreased by 0.003 or 0.3%. Moreover, The UNSW NB15 dataset experienced the same thing as the CICCIDS 2017 dataset, namely a small but significant decrease in accuracy. This is demonstrated by the accuracy obtained on the XGBoost model without SMOTE, which is 0.899, and the accuracy obtained after using SMOTE, which is 0.893, demonstrating that there is a decrease in accuracy but it is not too significant, namely equal to 0.006 or 0.6%. Based on the experimental results of testing the performance of the XGBoost model on the three IDS datasets mentioned above, it is demonstrated that SMOTE can improve the performance of the XGBoost model on the KDDCup99 dataset but not on the CICIDS 2017 or UNSW NB15 datasets.

The experimental results on the three IDS datasets show that the SMOTE method has no effect on the performance of the CatBoost model (Fig. 4). The accuracy of the CatBoost model obtained without SMOTE is 1,000 in the KDDCup99 dataset, and the accuracy obtained after using SMOTE is 0.999, demonstrating that there is a decrease in the performance of the CatBoost model, but only by 0.001 or 0.1%. The accuracy obtained without SMOTE in the 2017 CICIDS dataset was 0.997, and the accuracy obtained after using SMOTE was 0.996, indicating that there was a decrease in the performance of the CatBoost model, but not significantly by 0.001 or 0.1%. Furthermore, in the UNSW NB15 dataset, the accuracy obtained without SMOTE is 0.892, and the accuracy obtained after using SMOTE is 0.887, demonstrating that the performance of the CatBoost model has decreased but not significantly by 0.005 or 0.5%. Based on the results of the analysis of the experimental results of the CatBoost model on the three IDS datasets, it is demonstrated that the SMOTE method cannot significantly improve the performance of the CatBoost model.

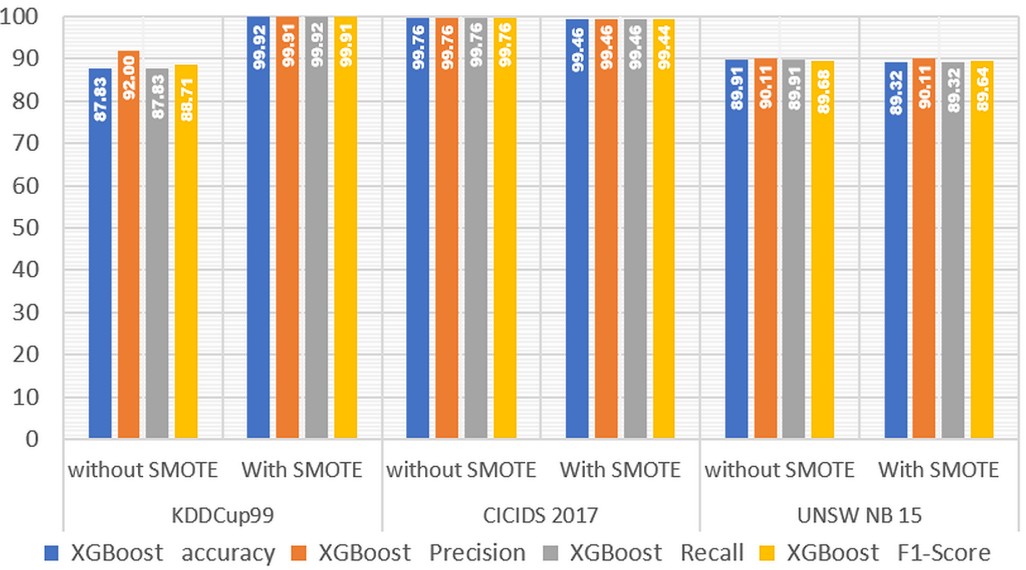

**Figure 3** XGBoost performance without and with SMOTE.

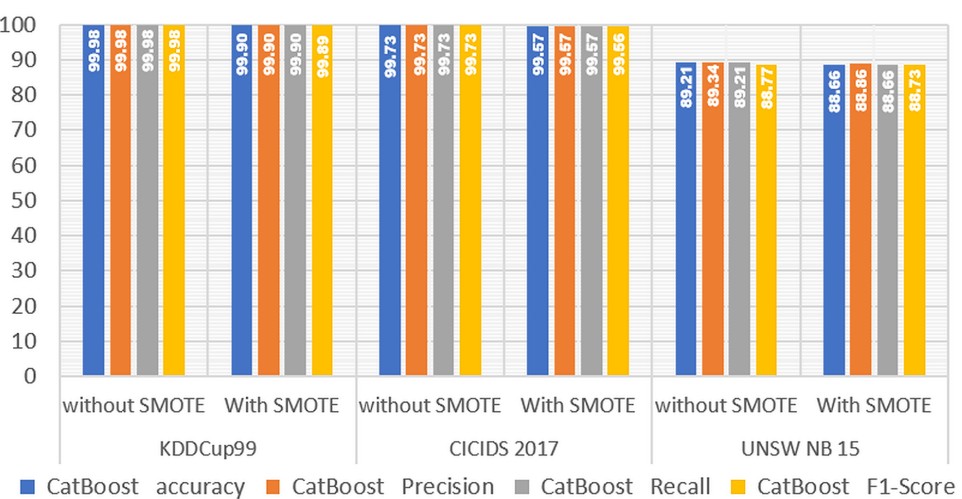

**Figure 4** CatBoost performance without and with SMOTE.

In addition to the performance in the previous discussion, in this experiment we tried to see the performance of time efficiency in carrying out the training process. Our experimental results are presented in Table 5.

Based on Table 5 and Fig. 5 it shows that in the KDDCup99 dataset, the computation time required for the LightGBM model without using SMOTE is 2.74 s, and the computation time required using SMOTE is 2.51, this proves that computing using SMOTE is 0.23 s faster. In CICIDS 2017 data, the computation time required for the LightGBM model without using SMOTE is 1.85, and the computation time required using SMOTE is 1.91,

**Table 5  Computing time.**

| Model | KDDCup99 | | CICIDS 2017 | | UNSW NB15 | |
|-------|----------|--------|-------------|--------|-----------|--------|
| | Without SMOTE | With SMOTE | Without SMOTE | With SMOTE | Without SMOTE | With SMOTE |
| LightGBM | 2.74 s | 2.51 s | 1.85 s | 1.91 s | 2.38 s | 2.43 s |
| XGBoost | 4.74 s | 21.1 s | 7.51 s | 7.8 s | 13.5 s | 14.7 s |
| CatBoost | 104 s | 52.7 s | 31 s | 31.2 s | 48.3 s | 49.8 s |

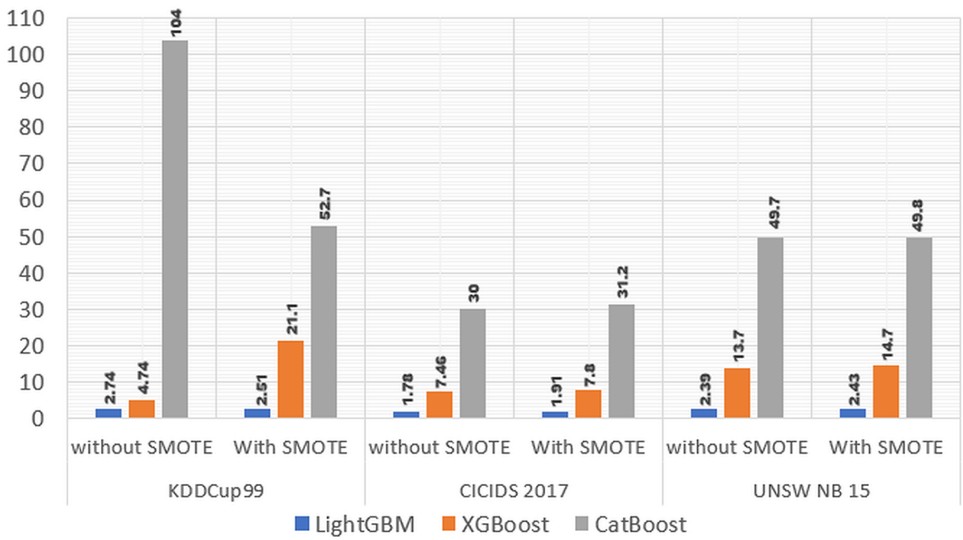

**Figure 5  Time consumed three models without and with SMOTE.**

this proves that computing using SMOTE is 0.13 s slower. Whereas in the UNSW NB15 data, the computation time required by the LightGBM model without SMOTE is 2.38, and the computation time using SMOTE is 2.43, this proves that computing using SMOTE is 0.04 s slower.

In the XGBoost model, when using the KDDCup99 dataset, the computation time required without using SMOTE is 4.74, and the computation time required using SMOTE is 21.1, this proves that the computation time using SMOTE is 16.36 s slower. When using the CICIDS 2017 dataset, the computation time required without using SMOTE is 7.51, and the computation time required using SMOTE is 7.8, this proves that the computation time using SMOTE is 0.34 s slower. When using the UNSW NB15 dataset, the computation time required without using SMOTE is 13.5, and the computation time required using SMOTE is 14.7, this proves that the computation time using SMOTE is 1 s slower.

In the CatBoost model, when using the KDDCup99 dataset, the computation time required without using SMOTE is 104 s, and the computation time required using SMOTE is 52.7 s, this proves that the computation time required using SMOTE is 51.3 s faster. When using the CICIDS 2017 dataset, the computation time required without using SMOTE is 31 s, and the computation time required using SMOTE is 31.2, this proves that

the computation time using SMOTE is 1.2 s slower. When using the UNSW NB15 dataset, the computation time required without using SMOTE is 48.3, and the computation time required using SMOTE is 49.8 s, this proves that the computation time using SMOTE is 0.1 s slower.

Based on the experimental results, it shows that the SMOTE method can improve the performance of the LightGBM model on the three IDS datasets, namely the KDDCup99, CICIDS 2017, and UNSW NB15 datasets. In the XGBoost model, in the KDDCup99 dataset, the SMOTE method was able to improve the performance of the XGBoost model, while in the other two datasets, namely CICIDS 2017 and UNSW NB15, the SMOTE method was not able to improve the performance of the XGBoost model but the difference between the method without SMOTE and using SMOTE was not too significant. In the CatBoost model, the experimental results show that SMOTE is not able to improve the performance of the three Boosting models on the three IDS datasets, but the difference between the three Boosting models without SMOTE and using SMOTE is not too significant.

Based on the analysis of the computational time required for the three Boosting models on the three IDS datasets, it shows that the average computation time required for the LightGBM model is 2.29 s, the average computation time required for the XGBoost model is 11.58 s, and the average computation time required for the CatBoost model is 52.9 s. This shows that the LightGBM model is faster compared to the XGBoost and CatBoost models. The computation time required for the three Boosting models on the three IDS datasets in the experimental results shows that the computation time without SMOTE is longer than the computation time using SMOTE except for the LightGBM and CatBoost models on the KDDCup99 dataset. This is because the amount of data processed by the three Boosting models using SMOTE on the three IDS datasets is greater than the processing of the three Boosting models on the three IDS datasets without using SMOTE.

## DISCUSSION

After all the processing is done, we compare our experimental results with some classic methods in the following lines: At the beginning of the first line, we compare them with naive Bayes and decision tree (*Ben Amor, Benferhat & Elouedi, 2004*), these two models are known for very low additional costs and high ones. The performance of these models is satisfactory. Also, Random Forest (*Zhang, Zulkernine & Haque, 2008*), a learning method that consists of many decision trees and is stronger in its generalization abilities than decision trees. Next comes the support vector machine (SVM) (*Wang, Wong & Miner, 2004*), a classic and efficient learning method, but it cannot handle big data (*Abdullah Alfrhan, Hamad Alhusain & Ulah Khan, 2020*). In addition, multilayer perceptron (MLP) (*Amato et al., 2017*) is a fundamental, classification-stable deep learning method. We also compare several class-balancing methods, such as random under sampling (RUS), random oversampling (ROS) (*Puri & Gupta, 2019*) and SMOTE (*Chawla et al., 2011*), which are identical in ensemble, with two methods, namely SVM and MLP, which produce samples concretely. All methods are divided into several sections, including RUS + MLP, RUS + SVM, ROS + SVM, SMOTE + SVM, ROS + MLP and SMOTE + MLP. We also compare our method to one of the learning convolution neural network (CNN)

**Table 6  Comparison results between three boosting and different methods (%).**

| Model | KDDCup99 | | UNSW NB15 | | CICIDS 2017 | |
|---|---|---|---|---|---|---|
| | Accuracy | F1 Score | Accuracy | F1 Score | Accuracy | F1 Score |
| Naïve Bayes | 73.55 | 72.31 | 61.8 | 65.27 | 93.90 | 93.53 |
| Decicision Tree | 77.89 | 75.25 | 73.25 | 76.36 | 99.62 | 99.57 |
| Random Forest | 77.20 | 73.23 | 74.35 | 77.28 | 99.79 | 99.78 |
| SVM | 72.85 | 68.84 | 68.49 | 70.13 | 96.97 | 96.99 |
| MLP | 78.97 | 75.40 | 78.32 | 76.98 | 99.48 | 99.39 |
| RUS + SVM | 73.57 | 70.11 | 67.16 | 70.45 | 96.45 | 96.55 |
| RUS + MLP | 76.66 | 72.38 | 77.27 | 76.21 | 99.46 | 99.42 |
| ROS + SVM | 73.34 | 69.90 | 68.32 | 70.00 | 96.98 | 97.04 |
| ROS + MLP | 78.10 | 74.18 | 76.13 | 76.97 | 99.55 | 99.55 |
| SMOTE + SVM | 79.23 | 78.36 | 71.5 | 73.77 | 97.00 | 97.04 |
| SMOTE + MLP | 77.47 | 75.18 | 79.59 | 80.10 | 99.33 | 99.34 |
| CNN | 78.33 | 74.75 | 80.52 | 76.61 | 99.48 | 99.44 |
| Fuzziness-based NN | 75.33 | 70.58 | 81.21 | 78.58 | 99.61 | 99.57 |
| LSSVM + MIFS ($\beta = 0.3$) | 78.20 | 72.76 | 76.83 | 77.43 | 98.76 | 98.67 |
| LSSVM + FMIFS | 75.67 | 73.67 | 77.18 | 77.65 | 99.51 | 99.48 |
| IGAN-IDS | 84.45 | 84.17 | 82.53 | 82.86 | 99.79 | 99.98 |
| SMOTE + LightGBM | 97.68 | 97.61 | 89.32 | 89.54 | 99.63 | 99.62 |
| SMOTE + XGBoost | 99.92 | 99.91 | 89.32 | 89.64 | 99.46 | 99.44 |
| SMOTE + CatBoost | 99.90 | 99.89 | 88.66 | 88.73 | 99.57 | 99.56 |

methods (*Li et al., 2017*). In addition, fuzziness-based neural network (NN) (*Ashfaq et al., 2017*), a semi-supervised learning approach method that can increase the generalization of IDS. Furthermore, our method is compared with Mutual Information Based Feature Selection (MIFS) before least square SVM (LSSVM) with an optimal value of = 0.3 (*Amiri et al., 2011*). Furthermore, we compare our proposed method with flexible MIFS (FMIFS) based on LSSVM + MIFS (*Ambusaidi et al., 2016*). Finally, our proposed method compares the IGAN-IDS (*Huang & Lei, 2020*) generalization method. all comparisons are shown in the Table 6.

From Table 6, Figs. 6 and 7, it can be seen that overall, the method we proposed has a higher level of accuracy and F1 score compared to other methods, except for the calculation of the CICIDS dataset, our proposed method is always even lower than the random and IGAN-IDS. This happens because the selection of the parameters is not entirely correct, so it is necessary to count on the selection of the appropriate parameters, so that the accuracy of the three models of shoes can be increased.

## CONCLUSIONS

We proposed STB (SMOTE on Tree Boosting) in this research to deal with imbalanced data sets on three IDS dates. Based on the method we propose, it has been proven to increase the capabilities of tree-boosting methods like LightGBM, XGBoost, and CatBoost. We used

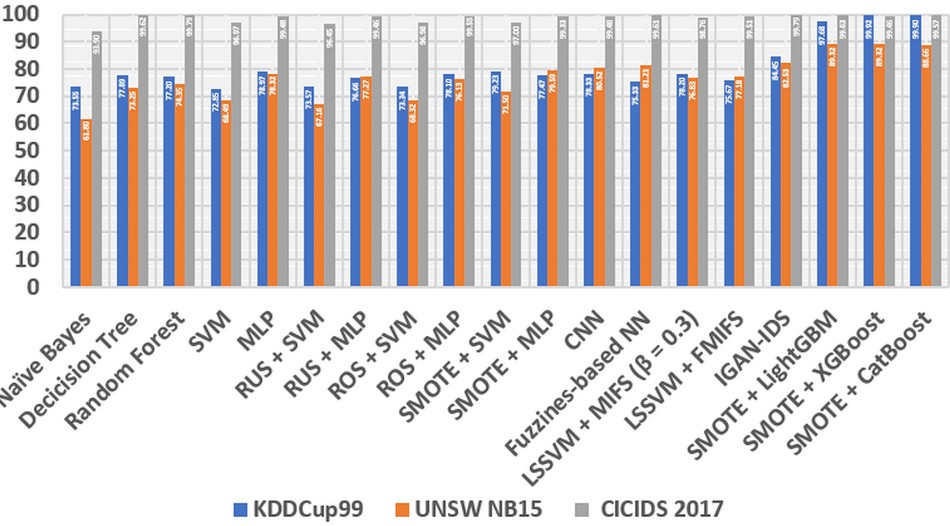

**Figure 6** Comparison STB and different methods based on accuracy.

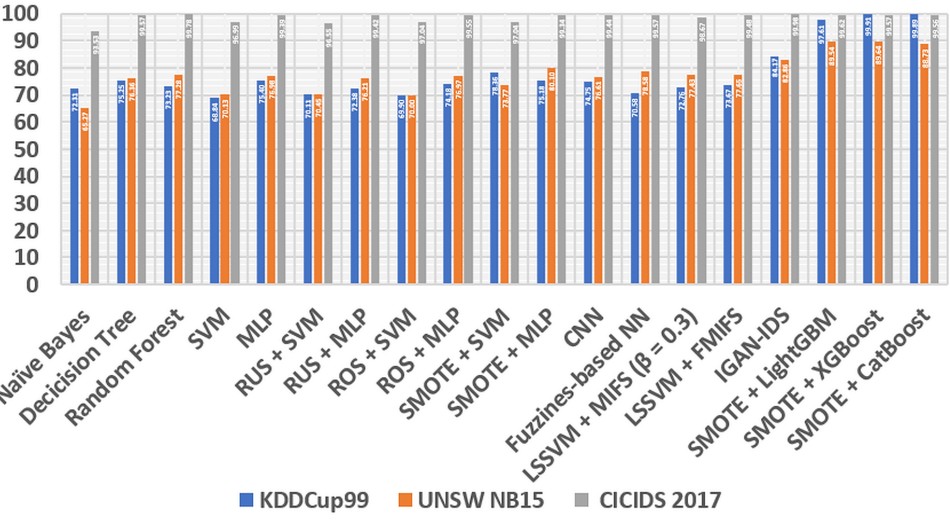

**Figure 7** Comparison STB and different methods based on F1 score.

three types of unbalanced IDS datasets, so this might be a good comparison to see how model tree enhancement works with and without SMOTE.

Using SMOTE also helps to better predict computational processes. proven by the value of the accuracy and the F1 score, which, with an average of 99%, is higher than several previous studies that tried to solve the same problem, namely imbalanced IDS dataset.

The analysis revealed that using SMOTE is beneficial for the LightGBM model in all three IDS datasets. For XGBoost, using SMOTE is good for KDDCup99, but requires re-experimentation with other methods, such as feature selection, for CICIDS 2017 and UNSW NB15 datasets. The use of SMOTE on the three boosting models and IDS data sets

for CatBoost needs to be improved by adding the feature selection method. These insights can help network administrators anticipate cyberattacks on IDS. To improve accuracy performance, future studies should supplement this method with future detection.

### Funding
The authors received no funding for this work.

### Competing Interests
The authors declare there are no competing interests.

### Author Contributions
- Li-Hua Li conceived and designed the experiments, analyzed the data, authored or reviewed drafts of the article, and approved the final draft.
- Ramli Ahmad conceived and designed the experiments, performed the experiments, performed the computation work, prepared figures and/or tables, and approved the final draft.
- Radius Tanone conceived and designed the experiments, performed the experiments, performed the computation work, prepared figures and/or tables, and approved the final draft.
- Alok Kumar Sharma performed the experiments, analyzed the data, authored or reviewed drafts of the article, and approved the final draft.

### Data Availability
The data is available at:

- KDD Cup 1999 Data: https://www.kaggle.com/datasets/galaxyh/kdd-cup-1999-data.

- Intrusion Detection Evaluation Dataset (CIC-IDS2017): https://www.unb.ca/cic/datasets/ids-2017.html.

- The UNSW-NB15 Dataset: https://research.unsw.edu.au/projects/unsw-nb15-dataset.

The code is available in the Supplemental File.

### Supplemental Information
Supplemental information for this article can be found online at http://dx.doi.org/10.7717/peerj-cs.1580#supplemental-information.

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
