# Peer review of "STB: synthetic minority oversampling technique for tree-boosting models for imbalanced datasets of intrusion detection systems"

_PeerJ Computer Science, doi:10.7717/peerj-cs.1580_

## Round 0.1 · original submission · Major Revisions

The reviewers have substantial concerns about this manuscript. The authors should provide point-to-point responses to address all the concerns and provide a revised manuscript with the revised parts marked via Tracked Changes

Reviewer 1 ·

Basic reporting

1 The review of related work is not sufficiently thorough and not sufficiently specific. The authors should cite the latest references and distinguish their works from others.

Experimental design

1 There are quite a few oversampling techniques, why did the authors only adopt SMOTE?
2 Since the data size is not big, why did the authors use the simple train-test split rather than other techniques such as cross-validation?
3 The authors should compare their methods to other published methods.
4 Statistical testing is missing.
5 How did the authors pick the optimized hyperparameter setting? Is there any comparison analysis?
6 There is lacking novelty in this paper. The authors should further improve it.

Validity of the findings

'no comment'

Additional comments

1 I suggest that the Discussion section should be improved to better reflect the quality of the work.

Reviewer 2 ·

Basic reporting

This paper provides a clear overview of the research topic and the proposed method. However, it would be helpful to include more specific details about the challenges of imbalanced datasets in intrusion detection systems. The use of SMOTE (Synthetic Minority Oversampling Technique) to generate synthetic tabular data for balancing imbalanced datasets is a relevant and well-established approach. However, it would be beneficial to provide a brief explanation of how SMOTE works and its advantages in this context.

The mention of using three boosting-based machine learning algorithms (LightGBM, XGBoost, and CatBoost) adds credibility to the study. However, it would be helpful to provide some rationale for choosing these specific algorithms and explain why they are suitable for this task.

The results indicating that using SMOTE improves the content accuracy of the LightGBM and XGBoost algorithms, as well as better predicting computational processes, are valuable findings. However, it would be beneficial to provide more specific details about the performance metrics used and how the improvement was measured.

The comparison of the proposed method with previous studies attempting to solve the imbalanced IDS dataset problem is important for demonstrating the effectiveness of the approach. However, it would be helpful to provide references to these previous studies and briefly discuss the differences in their methodologies.

The description of the proposed method in the "Proposed Method" section is clear and provides a step-by-step explanation of the workflow. Including a flowchart (Figure 1) enhances the understanding of the methodology.

The conclusions highlight the benefits of using SMOTE in improving the capabilities of tree-boosting methods and better predicting computational processes. However, it would be helpful to provide more insights into the specific challenges faced in each IDS dataset and discuss potential future directions for improvement.

Overall, the paper presents a novel method (STB) for dealing with imbalanced datasets in intrusion detection systems and provides valuable insights into the effectiveness of using SMOTE and boosting-based algorithms. However, some additional clarification and elaboration in certain sections would enhance the quality and impact of the paper.

Experimental design

no comment

Validity of the findings

no comment

Additional comments

Chinese characters should not appear in the paper. Typography and formatting need improvement.
The result display is too simple. In the experimental part, the form is used throughout.

---

## Round 0.2 · Major Revisions

The authors have addressed some of the concerns. However, one reviewer still has major concerns that concerns have not been fully addressed. The authors should address all the concerns (especially those from this reviewer) and make a point-to-point response to all concerns with revised parts being marked in different color.

Reviewer 1 ·

Basic reporting

None.

Experimental design

Most of the concerns in this part haven't been addressed well.

Validity of the findings

None.

Additional comments

None.

Reviewer 2 ·

Basic reporting

I appreciate authors’ efforts in conducting response to deal with my questions. The authors have basically addressed the points raised by this Reviewer in the revised version.
Recommendation: After formatted, considered for Acceptation.

Experimental design

no comment

Validity of the findings

no comment

---

## Round 0.3 · accepted · Accept

Reviewers are satisfied with the revisions and I concur to accept this manuscript.

Reviewer 1 ·

Basic reporting

None.

Experimental design

None.

Validity of the findings

None.

Additional comments

None.